# Convective Drying of Fresh and Frozen Raspberries and Change of Their Physical and Nutritive Properties

**DOI:** 10.3390/foods8070251

**Published:** 2019-07-11

**Authors:** Zoran Stamenković, Ivan Pavkov, Milivoj Radojčin, Aleksandra Tepić Horecki, Krstan Kešelj, Danijela Bursać Kovačević, Predrag Putnik

**Affiliations:** 1Faculty of Agriculture, University of Novi Sad, Trg Dositeja Obradovića 8, 21000 Novi Sad, Serbia; 2Faculty of Technology, University of Novi Sad, Bulevarcara Lazara 1, 21000 Novi Sad, Serbia; 3Faculty of Food Technology and Biotechnology, University of Zagreb, Pierottijeva 6, 10000 Zagreb, Croatia

**Keywords:** raspberry, convective drying, freeze-drying, bioactive compounds, shrinkage, color change

## Abstract

Raspberries are one of Serbia’s best-known and most widely exported fruits. Due to market fluctuation, producers are looking for ways to preserve this fresh product. Drying is a widely accepted method for preserving berries, as is the case with freeze-drying. Hence, the aim was to evaluate convective drying as an alternative to freeze-drying due to better accessibility, simplicity, and cost-effectiveness of Polana raspberries and compare it to a freeze-drying. Three factors were in experimental design: air temperature (60, 70, and 80 °C), air velocity (0,5 and 1,5 m·s^−1^), and state of a product (fresh and frozen). Success of drying was evaluated with several quality criteria: shrinkage (change of volume), color change, shape, content of L-ascorbic acid, total phenolic content, flavonoid content, anthocyanin content, and antioxidant activity. A considerable influence of convective drying on color changes was not observed, as ΔE was low for all samples. It was obvious that fresh raspberries had less physical changes than frozen ones. On average, convective drying reduced L–ascorbic acid content by 80.00–99.99%, but less than 60% for other biologically active compounds as compared to fresh raspberries. Convective dried Polana raspberry may be considered as a viable replacement for freeze-dried raspberries.

## 1. Introduction

Raspberries (Rubus idaeus) are one of the most important fruits in Serbian agriculture. It has been recently reported that raspberry production in 2017 was 109,742 t, which positions Serbia as one from the three leading countries in raspberry world production [1]. Over 90% of produced Serbian raspberries are commonly frozen, while only 10% is immediately used for processing or sold on open markets [2]. At their full maturity, raspberries have high moisture content (84% w.b.), L-ascorbic acid content, and potassium, as well as proteins, fibers, and minerals [3,4,5,6,7]; moreover, they contain various biologically active compounds (BACs) with high antioxidant activity. BACs found in raspberries include anthocyanins (cyanidin-3-sambubioside, cyanidin-3-glucoside, cyanidin-3-xylosylrutinoside, and cyanidin-3-rutinoside), ellagic acid, hydrolysable tannins (derivatives of gallic and ellagic acid), proanthocyanidins, vecetin, quercetin and rutin [8,9,10], carotenoids (lutein, zeaxanthin, alpha carotene, and beta carotene), chlorophyll derivatives, and tocopherols [11], with antioxidative and anti-inflammatory potential [12,13].

In Serbia, raspberries are often processed to jelly, diary, or confectionery products able to be stored for a longer time [6,14,15,16,17], however it is difficult to preserve this berries with high level of natural properties. In industry, this problem is commonly tackled with some form of drying, where one of the most popular procedures is convective drying that is done at high temperatures and for a long processing time. This often leads to chemical and biochemical changes and loss of native quality of this fruits with changes of color, taste, aroma, and nutritive value [18]. Previous studies showed that convective drying of red raspberries changed their volume, color, shape, and the content of BACs, this is especially true for contents of L–ascorbic acid, flavonoids, anthocyanins, and others [6,19,20,21]. Therefore, freeze-drying has been established as a standard process in the industry, to preserve nutritive value and extend the shelf life. A downside, however, is that freeze-drying is an expensive technology, not only in terms of the initial investment, but also during processing, even though previous reports showed that freeze-dried fruits, in comparison to convective dried ones, have better preserved their physiochemical, nutritive, and sensory properties [22,23]. The choice of the drying method for a particular food product is a crucial step as the drying procedure and operating conditions have impact on the quality of the dried product and its cost [24]. Despite its simplicity and low investment cost, convective drying is the most common dehydration technique in the food industry with a focus on minimizing economic and environmental impacts. Energy consumption and energy saving potential of convective drying may be tempered by combining new technologies with traditional drying procedures. Bórquez et al. [6] evaluated quality changes during osmotic dehydration of raspberries in sucrose solution with vacuum pretreatment, and followed by microwave-vacuum drying. Obtained results showed acceptable results regarding the preservation of color, taste and structure of these berries. Unfortunately, L-ascorbic acid content decreased 5-fold (220 to 41 mg/100 g), therefore reutilization of sucrose solution with higher initial concentration of L-ascorbic acid is recommended to reduce the losses to 60%.

Kowalski et al. [20] tested different drying techniques of raspberries, i.e., hot air drying vs. combined hybrid drying consisting of simultaneous hot air, microwave, and ultrasound drying, on the kinetics, energy consumption, and product quality. Results revealed that combined hybrid drying significantly improved the drying kinetics and the energy utilization. However, the total CIELab color difference (ΔE) ranged from 12 to 15, which implies significant degradation of color in dried raspberries. In addition, when final product was compared to fresh raspberry, considerable volumetric shrinkage was recorded with slight changes in the shape, but only after convective drying.

Bustos et al. [25] studied the impact of convective drying at various temperatures on phenolic characterization in both, raspberries (*Rubus idaeus* var. Autumn Bliss) and boysenberries (*Rubus ursinus × Rubus idaeus* var. Black Satin). Berries were applied to different convective drying conditions: 50 °C for 48 h, 65 °C for 20 h or 130 °C for 2 h until a moisture content was below 15%. Obtained results indicated that drying regime at 65 °C during 20 h was optimum for the best preservation of color, polyphenol content and antioxidant activity in dried berries. Moreover, authors argued that conventional drying is more economical than freeze-drying, and with considerable increase of total polyphenolic content due to depolymerization of native polyphenols which lead to improved antioxidant activity. Different raspberry cultivars, have different sensitivity to convective drying mostly evident by changes in their physical (e.g., reduced rehydration), mechanical (e.g., initial shape), nutritive (e.g., loss of nutrients), and sensory properties (e.g., formation of unpleasant aroma). Furthermore, Pavkov et al. [17] found that physiochemical properties and rehydration capacity of Polana and Polka varieties could be partially preserved after convective drying. 

Therefore, the aim of this research was to investigate the influence of convective drying on physiochemical properties and quality parameters of red Polana raspberry for fresh and frozen samples. Evaluated quality parameters included changes in volume and shape, color, L-ascorbic acid, total phenolic content, flavonoid content, anthocyanin content, and antioxidant activity. All results were controlled against freeze-dried raspberries.

## 2. Materials and Methods

### 2.1. Plant Material

Polana variety samples of red raspberries at full maturity were taken from the local farmers during August and October, 2017, at area of Novi Sad, Republic of Serbia. The raspberry fruits were harvested few hours prior to each experiment. The selected samples were approximately equal in size, volume, color, mass, and humidity. Average values of these properties were obtained from raspberry fruits subsample (*n* = 500) that were as following; (i) moisture content X_d.b_ = 5.45 ± 0.45 kg_w_/kg_d.b._; (ii) mass m = 3.30 ± 0.24 g; (iii) length a = 21.26 ± 0.74 mm, width b = 20.08 ± 0.56 mm, and thickness c = 18.63 ± 0.62 mm; (iv) volume V = 3.17 ± 0.22 cm^3^; and (iv) water activity a_w_ = 0.979 ± 0.001, at the temperature of 20 °C, total soluble solids of 6.6% and pH = 3.26 ± 0.02.

### 2.2. Convective Drying and Freeze-Drying of Raspberries

Raspberry fruits were dried as fresh or as frozen (with commercial freezer at −20 °C) with lab-scale convective dryer, designed and constructed to control for air flow (through the layer of processing material), air drying temperature, and continuous monitoring of the sample mass in processing [26]. The dryer chamber door was made up of glass, hence light could have some minimal impact on the quality drying products. In order to compare all results with controls, part of fresh fruits was freeze-dried with Martin Christ, Alpha 2-4 LDplus, without heating of the plates. Drying conditions were −83.8 °C on an ice condenser, with vacuum pressure of 0.0088 mbar in a drying chamber. Samples were dried for 48 h until average humidity at the end of the drying process of X_d.b_ = 0.07 kg_w_/kg_d.b_. 

### 2.3. Experimental Design and Statistical Analysis

Convective raspberry drying was three-factor experiment with one qualitative and two quantitative factors. The qualitative factor represented the initial state of the raspberry fruit before drying with two levels (fresh and frozen raspberries). Quantitative factors of the experiment were drying temperatures (60, 70, to 80 °C) and air velocity (0.5 to 1.5 m·s^−1^). The absolute air humidity was approximately constant in the experiments with average value 0.011 ± 0.002 kg_w_/kg_d.air_. For each experimental run, the initial mass of the raspberries was approximately 500 g. Raspberry samples were set on perforated sieve in a thin stagnant layer and placed in a drying chamber. Air flew along or across the surface of the material in the dryer, and the samples were dried until the same value of moisture content of approximately X_d.b_ = 0.152 kg_w_/kg_d.b._ The experiment was conducted with three repetitions for each experimental run.

### 2.4. Measuring of Volume and Shrinkage Determination

When drying some biomaterial, volume shrinkage (V_sh_) is one of the most common physical and quality indicators of the final product. Shrinkage is expressed by the ratio between the change of volume after drying and volume of sample before drying. The samples volume (*n* = 45) were measured by immersing the raspberries into 96% concentration of ethanol [27] according to
V_0_ = (m_0_ − m_l_)/ρ_t_(1)
where m_0_ is the mass of liquid and the immersed sample (kg), m_l_ is the mass of liquid, and ρ_t_ is the liquid density (kg/m^3^). The volumetric shrinkage of raspberries (V_sh_) was based on the following equation [28].
V_sh_ = ((V_0_ − V_i_)/V_0_) × 100(2)
where V_0_ is initial average volume and V_i_ is volume of each raspberry after drying.

### 2.5. Determination of Heywood Shape Factor

If observed independently, the volumetric shrinkage is not a sufficient indicator of the changes in dried material. For this reason, an additional indicator was used to monitor changes of shape, i.e., Heywood shape factor (k), able to detect the changes after drying [29,30,31]. This factor k = 0.523 and was calculated from the relation
k = V_p_/d_a_^3^(3)
where V_p_ is the particle volume with equivalent diameter of the projected area of the particles. This was obtained by assigning the area of an equivalent circle with the same greater diameter as that of the fruit [32].

### 2.6. Color Measurement and Total Color Difference

Before and after each drying regime, CIELab color parameters were assessed for raspberry samples (*n* = 45) by colorimeter Konica Minolta CR400(C-light source and the observer angle of 2°). Where L* was whiteness/brightness, a* was redness/greenness, and b* represented yellowness/blueness. The total color difference (ΔE), hue angle (h_o_) and chromaticity (C*) were calculated by Equations (4)–(6) [33,34]:ΔE = ((L* − L_0_)^2^+(a* − a_0_)^2^ + (b* − b_0_)^2^)^1/2^(4)
h_0_ = arctan (b*/a*)(5)
C* = ((a*)^2^ − (b*)^2^)^1/2^(6)
where, L_0_, a_0_, and b_0_ are the color values before drying, while L*, a*, and b* are the color values after drying.

### 2.7. Analysis of Nutritiveproperties

#### 2.7.1. Extraction Procedure

Methanol extract was prepared from the fresh/dried raspberry samples to determine the contents of total phenols, flavonoids, and radical scavenging capacity. Briefly, 50 mL of an extraction solvent (methanol, 99.8%) (Fisher Scientific, UK) was poured over the raspberry sample into an Erlenmeyer flask. The flasks were covered and placed on a laboratory stirrer for 24 h (in a dark place). After the extraction, the samples were transferred to volumetric flasks of determined volume, filtered, and stored in a dark and cool place until the analysis was carried out.

#### 2.7.2. Determination of Total Phenolic Content (TPC)

The TPC in methanol extracts of fresh and dried raspberries was determined by Folin–Ciocalteu spectrophotometric method [35]. In a 50 mL volumetric flask with V = 0.5 mL of extract, 0.25 mL of Folin–Ciocalteu was mixed and 0.75 mL of 20% Na_2_CO_3_ (*m*/*v*). After 3 min of stirring, distilled water was added and made up to volume of 50 mL. Reaction mixture was left to stand at room temperature for 2h and absorbencies were measured at 765 nm by UV–Vis spectrophotometer. Based on the measured absorbance, the concentration (mg/mL) of TPC was calculated from the calibration curve of the standard solution of gallic acid. The results are expressed in g of gallic acid equivalents (GAE) per 100 g of fruit dried basis (gGAE/100g_d.b_).

#### 2.7.3. Determination of Total Flavonoids Content (TFL)

The TFL was determined by previously described colorimetric method [36]. In short, the reaction mixture was prepared by mixing 1 mL of an extract with 4 mL of distilled water and 0.3 mL of a 5% NaNO_2_ solution (*m*/*v*). Then mixture was incubated at room temperature for five minutes, and then 0.3 mL of 10% AlCl_3_ (*m*/*v*) was added. After six minutes, when the solution became very yellow, 2 mL of NaOH was added. Distilled water was added to the reaction mixture and made up the volume to 10 mL in a volumetric flask. The absorbance was measured at 510 nm. The TF were calculated according to the catechin standard calibration curve and expressed in mg of catechin equivalents (CAE) per 100 g of fruit dried basis (mgCAE/100g_d.b_).

#### 2.7.4. Determination of Radical Scavenging Capacity

The free radical scavenging capacity (RSC) of raspberry extracts was determined using a simple and fast spectrophotometric method described by Espin et al. [37]. Briefly, the prepared extracts were mixed with methanol (95%) and 90 μM 2,2-diphenyl-1-picryl-hydrazyl (DPPH) to give different final concentrations of extract. After 60 min at room temperature, the absorbance was measured at 517 nm. RSC was calculated according to Equation (7) and expressed as IC50 value, which represents the concentration of extract solution required for obtaining 50% of RSC.
RSC (%) = 100 − (A_sample_ × 100)/A_blank_(7)
where A_blank_ is the absorbance of the blank and A_sample_ is the absorbance of the sample. The obtained results were presented as a mass of dry sample material that is necessary for inhibition of 50% of DPPH (IC50 (mg_d.b_/mL)). 

#### 2.7.5. Determination of Monomeric Anthocyanin Content (AC)

The sample preparation for the content of total AC was conducted by previously described method [38]. Here an extraction solvent (ethanolic acid solution) [39] was poured over the samples (fresh or dried raspberries) and the mixture was thoroughly homogenized in a glass beaker. Afterwards, the beaker was covered with paraffin film and left to sit at 4 °C. After 24 h, the extraction mixture was kept at room temperature, filtered, and transferred to volumetric flask and made up to the volume of 100 mL with an extraction solvent. An aliquot of an extract was transferred into two volumetric flasks with added buffers at pH = 1.0 and pH = 4.5. After 15 min, the absorbencies were measured at 510 and 700 nm against distilled water as a blank. The content of AC was recalculated to cyanidin-3-glucoside by
AC = (A × M_w_ × D_f_ × V_m_ × 1000)/(ɛ × m)(8)
A = (A_510_ − A_700_)pH_1,0_ − (A_510_ − A _700_)pH_4,5_(9)
where AC = anthocyanin content (mg/100g); A_510_ = sample absorbance at λ = 510nm; A_700_ = sample absorbance at λ = 700nm; M_w_ = molecular weight of cyanidin-3-glucoside (449.2), D_f_ = dilution factor = original solution volume; ɛ = molar extraction coefficient of cyanidin-3-glucoside (26900); and m = sample weight (g). The content of AC was expressed in mg per 100 g of fruit dried basis (mg/100g_d.b_).

#### 2.7.6. Determination of Vitamin C Content 

Separations and quantifications of vitamin C were performed by HPLC equipment (Thermo Scientific™ UltiMate 3000) on Nucleosil 100-5C_18_, 5 µm (250 × 4.6 mm I.D.) column (Phenomenex, Los Angeles, CA). Separation was performed with standard method BS EN 14130:2003 (Foodstuffs. Determination of vitamin C by HPLC). The content of AC was expressed in mg ascorbic acid (the sum of ascorbic acid and its oxidative form of dehydroascorbic acid) per 100 g of fruit dried basis (mg/100g_d.b_).

### 2.8. Statistical Analysis

For the purposes of statistical tests, analysis of variance was performed (ANOVA) with Statistica13 (Stat Soft, Inc., Oklahoma, United States). In order to define homogenous groups of samples an additional Duncan test was performed with statistical significance at *p* < 0.05.

## 3. Results and Discussion

### 3.1. Volume Shrinkage

Comparison of convective and freeze-drying technique for fresh vs. frozen samples revealed that the least changes in volume had freeze-drying (V_sh_ = 16.49 ± 2.75%). For convective drying, the least changes in volume had fresh raspberry samples dried at T = 60 °C and air velocity of 1.5 m·s^−1^ (V_sh_ = 35.74 ± 6.78%). Pavkov et al. 2017 [17] reported results of air drying red Polana raspberry, dried at air temperature of 50, 60, 70, and 80 °C and constant air velocity of 1 m·s^−1^. Judging by the volume shrinkage during convective drying, air temperature of 50 °C will lead to a total collapse and loss of the product shape. Interestingly, the least volume shrinkage (23.17%) was achieved with air temperature at 70 °C. Drying with air temperature at 60 °C provoked shrinkage of 28.74%, what is still lower than it was obtained in the current study. Samples dried with air temperature of 80 °C reached volume shrinkage of 43.13%. Results obtained from these two experiments revealed that higher air temperatures do not necessarily lead to higher changes in volume shrinkage. This may be explained by the fact that higher air temperature have tendency to lean towards mechanical stabilization of the raspberry surface, thus limiting the degree of shrinkage.

Air temperature and initial state of the raspberry, prior to convective drying significantly changed the volume of dried raspberry. On the other hand, the air velocity did not have any impact on the change in raspberry volume (Table 1). The most considerable changes in volume occurred when drying frozen raspberries at T = 80 °C and air velocity of 0.5 m·s^−1^ (V_sh_ = 79.07 ± 4.07%). As previously reported, drying temperature of 80 °C had similar trend on volume shrinkage [17].

Additional for convective drying, some reports indicated variations in volume shrinkages with regards to raspberry varieties. Sette et al. [29] dehydrated with convective drying previously frozen Autumn Bliss raspberry, at T = 60 °C and air velocity of 1–1.5 m·s^−1^. Here they found higher volume shrinkage (V_sh_ = 81 ± 3%) than what was reported in the current study.

Initial raspberry state effected the shrinkage, which was not surprising as creation of the ice crystals tends to destabilize cellular structures and this is particularly emphasized with drying air velocity of 0.5 m·s^−1^. On the contrary, Duncan’s test revealed that there are no significant differences in the volume shrinkage of the samples which were dried at the same temperature and with velocity of1.5 m·s^−1^. The reason for this may be the faster drying rate in the first drying period, which can lead to a faster mechanical stabilization of the surface, hence the preservation of the volume. As expected, initial raspberry state had an effect on volume shrinkage, and the results after drying are presented in Figure 1. Figure 2 shows the shrinkage and the changes in fruit size after convective and freeze-drying of fresh raspberry at T = 60 °C and air velocity of 1.5 m·s^−1^.

### 3.2. Heywood Shape Factor Results

The referent Heywood shape factor before drying of fresh raspberry was k = 0.3323 (Figure 3), and all three experimental factors were significant for the changes in Heywood shape factor. As compared to k of a fresh raspberry, the factor after freeze-drying equaled to k = 0.27. In case of convective drying, the lowest deviation from k occurred for fresh raspberries at T = 60 °C and air velocity of 1.5 m·s^−1^ (k = 0.2694 ± 0.003). Results showed that under the same experimental conditions, the convective dried frozen raspberries had greater deviation of size as compared to the fresh raspberries. Hence, Heywood shape factor corresponded with the results for volume shrinkage.

### 3.3. Color Change

CIE Lab color parameters L*, a*, b*, C*, h*, and ΔE* measured on fresh and dried raspberries at different drying conditions are shown in Table 2. Any considerable influences on color caused by the drying of raspberry was not detected, as total color change was roughly 10, except for freeze dried samples, and essentially lightness (L) remained similar to those of a fresh samples. Hence, changes in color were driven by the parameters a* and b*. Generally, convective dried raspberry samples slightly shifted towards maroon color, which can originate from decomposition of carotenoid pigments. Moreover, high temperature induces nonenzymatic Maillard browning with formation of brownish pigmentations [40]. Alternatively, this may be the consequences of high concentrations of preserved anthocyanins in dried samples [3]. A slight increase of a* and b* will have positive repercussions, as it will lean towards more saturated color of products, which corresponds well with increased chroma values.

Air temperature, air velocity, and initial state of raspberry before drying had statistically significant effect on color change (*p* < 0.05) (Table 1). The least color change (ΔE = 5.18) was observed with convective drying at T = 60 °C and air velocity of 1.5 m·s^−1^. This temperature remained optimal choice regarding ΔE, as it was not modified by different air velocities and initial state of material (fresh and frozen). The largest color change for this drying type was at T = 80 °C and air velocity of 0.5 m·s^−1^ for both frozen and fresh raspberry when ΔE was 11.17 and 10.12, respectively. Unexpectedly, freeze-dried raspberries had the largest color changes (ΔE = 19.60) that were caused by increase in all of the three-color parameters, and especially for a* (Δa = 18.04). Bustos et al. [25] reported similar findings for freeze-dried berries with higher values for redness (Δa = 25.62) as compared to convective samples. Also, study by Sette et al. [3] reported an increase of a* and emphasized that besides pigmentation, differences of internal structures should be considered among convective and freeze dried raspberries. For instance, after freeze-drying, free water from raspberry is replaced by air, so shifts in red color and lightness can be a consequence of different diffusion of light that passes throughout a material. This effect is likely more pronounced for fruits with defined and vibrant hues, as for the raspberries [3,41]. During the conventional air-drying, increasing drying temperatures reduce the drying time, whereas shorter drying times may result in reduced risks of food quality deterioration [23]. Increasing hot air temperature for convective drying of *Cassia alata* from 40 °C to 60 °C reduced drying time from 180 min to 120 min [42]. Consequently, from data obtained, it can be assumed that as the temperature and the drying time increase, the color change of dried raspberries will increased too.

### 3.4. Ascorbic Acid Reduction

The average amount of L-ascorbic acid in fresh samples before drying was 118.27 mg/100g_d.b._ (18.92 mg/100g_w.b._) (Table 4), which was similar to quantities reported by Bobinaite et al. [12]. This content of L-ascorbic acid was significantly reduced during convective drying under all experimental conditions. This is expected as prolonged exposure to heightened temperatures and oxygen has tendency to reduce the content of this acidin fruits [21,43,44,45,46,47]. Figure 4 shows temperature kinetics of raspberry samples during convective drying from a fresh state. Type K thermocouple probes were used to monitor and control product temperature during the process, by placing probes inside the drupelet. For all experiments, the temperature at the beginning of the process is approximately 35 °C, but after 10 minutes of the drying, the raspberry temperature can reach 50 °C. Due to the reduced moisture content, during the last quarter of drying all samples have the same temperature as drying air. This means that drying time at T = 80 °C is 6–8 h, and depending of the air velocity can last almost three times longer at T = 60 °C. However, the highest content of L-ascorbic acid was after the shortest convective drying with air velocity of 1.5 m·s^−1^. This was regardless of the fact that the raspberry temperature reached T = 80 °C, and equaled to 27.46 ± 1.12 mg/100g_d.b._ and 22.54 ± 1.28 mg/100_d.b._, after drying of frozen and fresh raspberry, respectively (Table 4). Conversely, the degradation of 99% L-ascorbic acid was detected for longest drying with T = 60 °C and air velocity of 0.5 m·s^−1^. Accordingly, this might mean that L-ascorbic acid degradation is more induced by longer exposure to higher oxygen levels during the convective drying than to the drying temperature itself.

This reasoning is in accordance with Verbeyst et al. [43] research with thermal and high-pressure effects on vitamin C degradation in strawberries and raspberries. Here it was shown that ascorbic acid degradation from strawberry and raspberry is slightly temperature dependent for temperature range of 80 to 90 °C, and that oxygen presence plays the key role. As expected, the highest levels of L-ascorbic acid preservation was achieved by freeze-drying (115.48 ± 2.29 mg/100g_d.b_.), since there was neither thermal nor oxygen degradation involved.

### 3.5. Total Phenols Reduction

The average values for relevant nutritive profile of fresh raspberries are presented in Table 3. Average total amount of polyphenols in fresh raspberry was 1.63 g GAE/100g_d.b_. All three individual experimental factors had influence on the content of total phenols (Table 1).

When these samples were dried convectively the best preserved polyphenolic content was at T = 70 °C and air velocity of 1.5 m·s^−1^ (1.28 gGAE/100g_d.b_.). On the contrary, they were least preserved at 60 °C and air velocity of 0.5 m·s^−1^(0.92 gGAE/100g_d.b_.). Freeze-drying preserved 1.10 g GAE/100g_d.b_. of total phenols, and, as expected, convective drying reduced polyphenolic content in the samples. The exceptions were the samples freshly dried at air velocity of 1.5 m·s^−1^ and drying temperatures of 70 °C; and 80 °C in which higher total phenolic content was observed in comparison to freeze-dried samples. Similar results were recently reported where higher phenolic content was found in convectively hot air-dried *Cassia alata* in comparison to freeze-dried samples [42]. Hossain et al. has suggested that freeze-drying may not have completely deactivated degradative enzymes due to the low-temperature process. Therefore, reactivation of this degradative enzymes could be further occurred in freeze-dried samples thus result in lower phenolic content [48].

Vasco et al. [49] made classification of 17 fruit types from Ecuador based on their content of total phenols and according to this classification there are three main groups: one with low levels of total phenols (<0.1 gGAE/100g_w.b._), one with medium level (0.2–0.5 gGAE/100g_w.b._), and the third with high levels (>1.0 gGAE/g100g_w.b._). This classification was accepted by others [50,51], and states that fresh Polana raspberry belongs to a high content group, as do freeze-dried and convectively dried samples from this study (under all experimental conditions).

### 3.6. Total Anthocyanin Reduction

Temperature had significant influence on the content of anthocyanin (Table 4), however air velocity had no effect on this group of compounds. Amount of anthocyanin in fresh raspberry was 511.7 mg/100g_d.b._ After convective drying, anthocyanin content was preserved from 40–56%. Anthocyanin content (287.0 mg/100_d.b._) was best preserved with drying of fresh raspberries at T = 70 °C and air velocity of 0.5 m·s^−1^. Their least retention occurred after convective drying of frozen samples at T = 70 °C and air velocity of 0.5 m·s^−1^ (205.3 mg/100_d.b._). Thermal degradation of anthocyanins and complementary oxidization is the origin of the maroon color that was detected with the CIELab analysis. After freeze-drying, the content of anthocyanin in raspberry was 410.4 mg/100g_d.b._ which is expected due to minimized considerable influence of temperature and oxygen. 

### 3.7. Radical Scavenging Capacity

As expected, all experimental factors influenced the radical scavenging capacity (Table 4). As a smaller IC50 means higher radical scavenging capacity, the majority of convectively dried raspberries exhibited lower radical scavenging capacity in comparison to fresh or freeze-dried samples. The IC50 value of fresh raspberry was IC50 = 0.0534 mg_d.b._/mL. Freeze-dried samples had highly preserved radical scavenging capacity that was equal to 0.0641 mg_d.b._/mL, likely due to high preservation of all bioactive compounds. The lowest IC50 value (e.g. the highest radical scavenging capacity) had convective drying for frozen samples, of IC50 = 0.0845 mg_d.b._/mL which was obtained at T = 80 °C and air velocity of 0.5 m·s^−1^. The main reason for this may be the high preservation of total flavonoid content (0.97%) in dried samples with same convective drying regime. Raspberry belongs to a group of biomaterial with high radical scavenging capacity [12,50]. It is also believed that almost 20% of its total radical scavenging capacity is secured by the content of L-ascorbic acid [52]. As previously reported, heat and oxygen have influence on almost all bioactive compounds with some form of degradation, so it is not surprising that to find the loss of radical scavenging capacity due to convective drying.

## 4. Conclusions

Using physical properties, contents of various biologically active compounds and radical scavenging capacity proved to be useful in selecting alternatives for preservation of raspberries as in the case of convective and freeze-drying. For Polana variety, the most desirable results against freeze-drying as standard in terms of color, volume shrinkage, and Heywood shape factor change was achieved with convective drying of fresh raspberry at T = 60 °C with air velocity of 1.5 m·s^−1^. Convective drying of raspberry had influenced all measured biologically active compounds. In comparison to fresh samples, in convectively dried raspberries 60–78% of total phenols was preserved as well as 75–97% of flavonoids and 40–56% of anthocyanins. Consequently, lower radical scavenging capacity was found in convectively dries samples as compared to fresh or freeze-dried. The largest shortcoming for convective drying was observed in difference between freeze-dried for preservation levels of L-ascorbic acid. Freeze-drying preserved more than 97% of L–ascorbic acid, while convective drying samples had degradation of over 80% of this compound. This might not be as relevant where L–ascorbic acid is added in processing of raspberries (e.g., confectionery products, biscuits, cookies, dairy product etc.). In conclusion, Polana raspberry dried convectively with air temperature of 60 °C and air velocity of 1.5 m·s^−1^, may be considered as sufficient alternative to freeze-drying.

## Figures and Tables

**Figure 1 foods-08-00251-f001:**
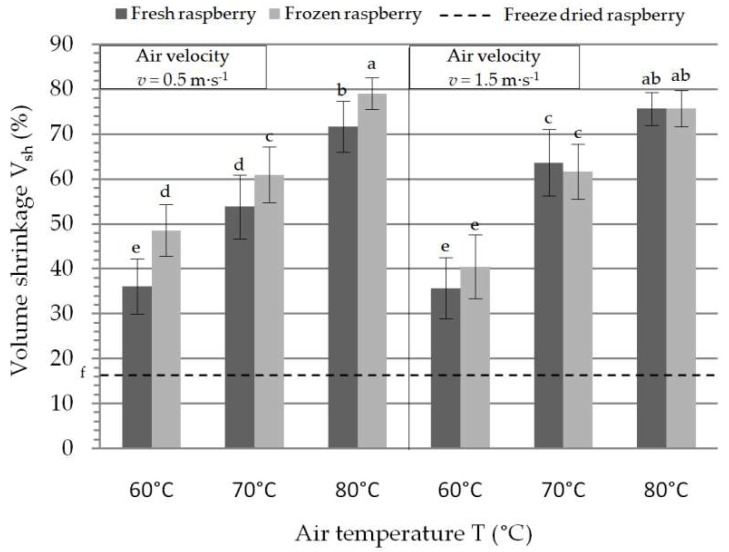
Volume shrinkage after convective drying of raspberry variety Polana. Different lowercase letters indicate significant differences (*p* < 0.05).

**Figure 2 foods-08-00251-f002:**
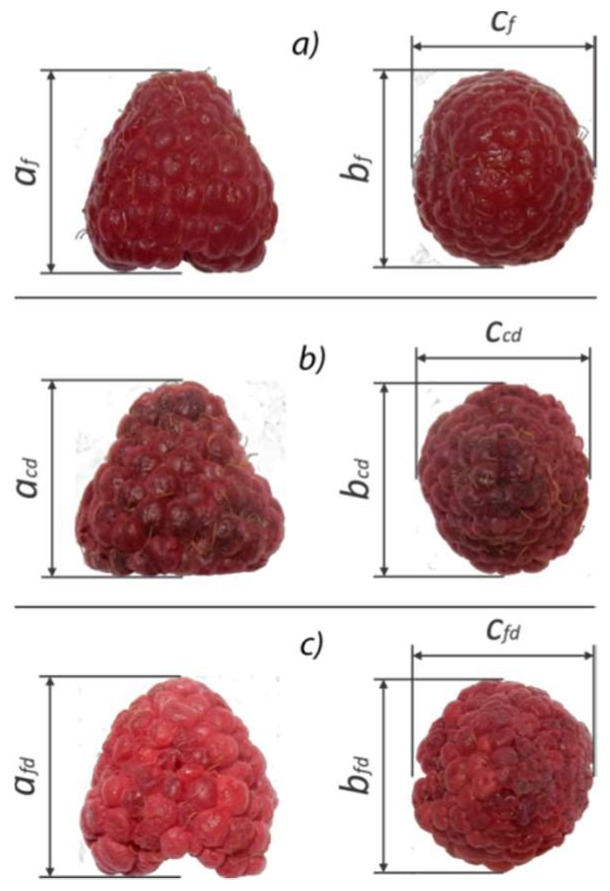
Red raspberry, variety Polana: (**a**) fresh (a_f_ = 20.74±2.45mm; b_f_ = 20.04 ± 1.96 mm; c_f_ = 18.88 ± 1.72 mm), (**b**) after convective drying of fresh raspberry at T = 60 °C and air velocity of 1.5 m·s^−1^(a_cd_ = 16.97 ± 1.86 mm; b_cd_ = 15.37 ± 1.65 mm; c_cd_ = 15.34 ± 1.43 mm), and (**c**) after freeze-drying (a_fd_ = 20.62 ± 1.98 mm; b_fd_ = 20.62 ± 2.66 mm; c_fd_ = 18.58 ± 1.78 mm).

**Figure 3 foods-08-00251-f003:**
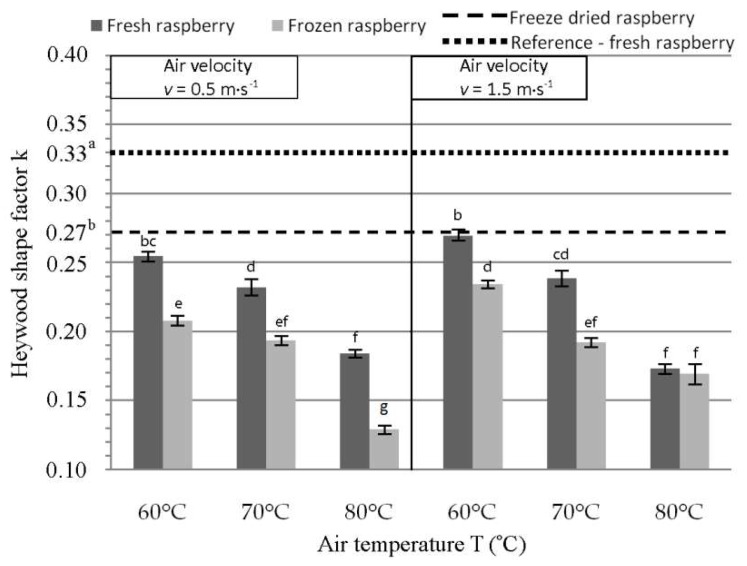
Of Heywood shape factor after convective drying of Polana raspberry. Different lowercase letters indicate significant differences (*p* < 0.05).

**Figure 4 foods-08-00251-f004:**
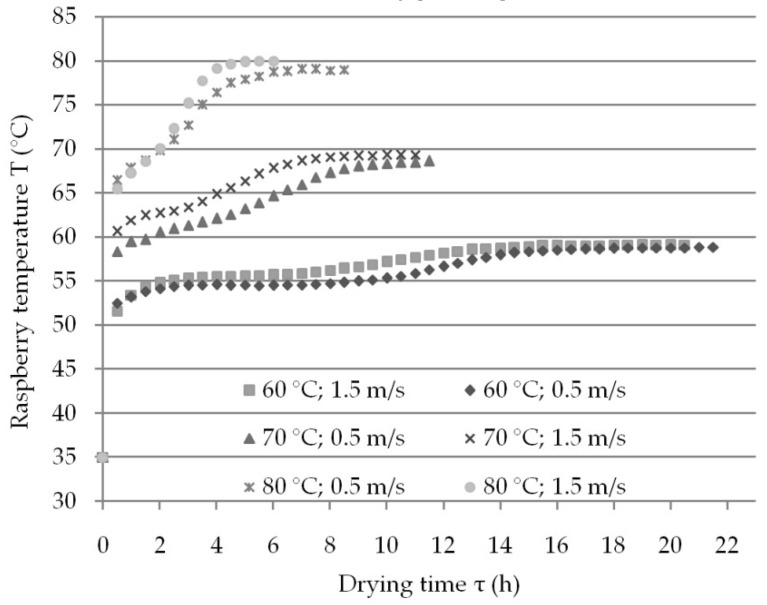
Raspberry temperature kinetics during convective drying from a previously fresh state.

**Table 1 foods-08-00251-t001:** Statistical analysis of convective drying factor effect on all determined quality indicators.

Depend. Value	Statistical Indicators	Effect								
		Intercept	AT	AV	RS	AT*AV	AT*RS	AV*RS	AT*AV*RS	Error
Volume shrinkage	SS	1687039	96255	11	3834	2874	1686	2743	103	28630
	MS	1687039	48127	11	3834	1437	843	2743	51	59
	F	28814.82	822.02	0.20	65.49	24.54	14.40	46.85	0.88	
	p	0.0000	0.0000	0.6587	0.0000	0.0000	0.0000	0.0000	0.4166	
Heywood shape factor	SS	0.208363	0.0044	0.0001	0.0015	0.0000	0.0000	0.0001	0.0001	0.0094
	MS	0.208363	0.0022	0.0001	0.0015	0.0000	0.0000	0.0001	0.0000	0.0000
	F	10773.36	115.30	10.08	79.20	1.59	1.16	7.79	4.55	
	p	0.0000	0.0000	0.0015	0.0000	0.2057	0.3129	0.0054	0.0110	
Color change	SS	17759.28	576.41	16.78	68.15	31.92	56.31	0.57	1.77	654.85
	MS	17759.28	288.20	16.78	68.15	15.96	28.15	0.57	0.89	2.74
	F	6481.604	105.186	6.126	24.871	5.825	10.275	0.207	0.323	
	p	0.000000	0.000000	0.014017	0.000001	0.00330	0.000052	0.649312	0.724239	
Ascorbic acid	SS	1689.955	1357.504	640.217	6.840	430.577	91.156	0.008	7.249	8.575
	MS	1689.955	678.752	640.217	6.840	215.289	45.578	0.008	3.625	0.357
	F	4729.853	1899.694	1791.841	19.144	602.551	127.564	0.024	10.144	
	p	0.000000	0.000000	0.000000	0.000203	0.000000	0.000000	0.879336	0.000641	
Total phenolic content	SS	40829272	96596	58924	52597	8089	23366	65802	46258	10129
	MS	40829272	48298	58924	52597	4044	11683	65802	23129	422
	F	96744.51	114.44	139.62	124.63	9.58	27.68	155.92	54.80	
	p	0.0000	0.0000	0.0000	0.0000	0.0008	0.0000	0.0000	0.0000	
Total flavonoid content	SS	3799457	6538	2505	1149	8042	1435	6	4362	5100
	MS	3799457	3269	2505	1149	4021	718	6	2181	213
	F	17879.73	15.38	11.79	5.40	18.92	3.38	0.03	10.26	
	p	0.000000	0.000050	0.002170	0.028864	0.000012	0.051007	0.863581	0.000602	
Anthocyanin content	SS	2163673	22925	715	2014	711	2315	782	9184	18331
	MS	2163673	11463	715	2014	355	1157	782	4592	764
	F	2832.821	15.008	0.936	2.637	0.465	1.515	1.024	6.012	
	p	0.0000	0.0000	0.3428	0.117456	0.6334	0.2400	0.3215	0.0076	
Radical scavenging	SS	0.696911	0.003357	0.000115	0.011585	0.027292	0.008212	0.009029	0.015719	0.00044
	MS	0.696911	0.001678	0.000115	0.011585	0.013646	0.004106	0.009029	0.007860	0.00001
	F	37390.89	90.05	6.18	621.55	732.14	220.29	484.40	421.69	
	p	0.000000	0.000000	0.020259	0.000000	0.000000	0.000000	0.000000	0.000000	

AT—Air Temperature; AV—Air Velocity; RS—Raspberry state before drying; SS—Sum of Squares; MS—Mean square; F—Fisher’s F-ratio; p— *p*-value.

**Table 2 foods-08-00251-t002:** Results of CIE Lab parameters before and after all drying treatments: L* (whiteness/brightness), a* (redness/greenness), b* (yellowness/blueness), and ΔE (color change).

Experiment Factors					Measured Values										ΔE
				Before Drying (Fresh Samples)						After Drying (Dried Samples)					
				L_0_	a_0_	b_0_	h_0_^o^	C_0_*		L*	a*	b*	h^o^	C*	
Fresh dried	0.5 m·s^−1^	Drying air temperature [°C]	60	24.829	22.172	11.091	26.575	24.791		23.848	27.570	11.924	23.388	30.038	5.549 ± 1.67 ^h^
			70	24.768	20.932	10.134	25.833	23.256		25.799	29.650	13.448	24.397	32.557	9.383 ± 1.66 ^def^
			80	26.225	22.251	9.5562	23.242	24.216		22.608	30.589	14.021	24.625	33.649	10.12 ± 1.10 ^bc^
	1.5 m·s^−1^		60	24.407	22.009	10.553	25.617	24.408		23.058	26.982	11.101	22.363	29.176	5.181 ± 0.95 ^h^
			70	24.372	22.513	10.886	25.805	25.006		24.194	30.224	13.100	23.433	32.940	8.024 ± 1.16 ^efg^
			80	25.212	22.125	10.084	24.502	24.314		22.029	30.390	14.193	25.033	33.540	9.763 ± 1.21 ^cde^
Frozen dried	0.5 m·s^−1^		60	24.491	21.032	10.397	26.305	23.461		24.264	28.325	12.834	24.375	31.096	7.692 ± 1.23 ^g^
			70	24.078	20.509	9.414	24.655	22.566		24.094	29.314	12.897	23.747	32.025	9.468 ± 1.62 ^def^
			80	26.368	22.380	10.172	24.442	24.583		26.329	32.674	14.529	23.973	35.758	11.17 ± 1.38 ^b^
	1.5 m·s^−1^		60	25.147	21.106	9.865	25.051	23.297		24.514	28.904	12.892	24.038	31.648	8.38 ± 1.43 ^fg^
			70	25.122	21.477	10.321	25.667	23.828		23.603	29.665	13.733	24.841	32.689	8.99 ± 0.80 ^def^
			80	25.785	22.719	10.593	24.997	25.067		21.991	30.368	13.977	24.714	33.430	9.18 ± 1.27 ^cd^
Freeze dried				24.557	22.367	9.779	23.615	24.411		28.144	40.412	16.562	22.285	43.674	19.608 ± 1.63 ^a^

Different lowercase letters indicate significant differences (*p* < 0.05).

**Table 3 foods-08-00251-t003:** The content of BACs and radical scavenging capacityin fresh raspberry sample.

Compound	mg/100g_d.b._
L-ascorbic acid	118.27
Total phenolic content	1,635.60
Flavonoid content	386.19
Anthocyanin content	513.55
Antioxidative activity IC50 [mg_d.b._/mL]	0.0534

**Table 4 foods-08-00251-t004:** Results of biologically active compounds in fresh raspberry and dried raspberry after all drying treatments.

Experimental Factors				Measured Values				
				Ascorbic Acid (mg/100g_d.b._)	Total Phenolic Content (gGAE/100g_d.b._)	Total Flavonoid Content (mgCAE/100g_d.b._)	Anthocyanin Content (mg/100g_d.b._)	Radical Scavenging IC50 (mg_d.b._/Ml)
Fresh dried	0.5 m·s^−1^	Drying air temperature (°C)	60	<0.25 ^d^	0.921 ± 0.010 ^k^	315.1 ± 8.9 ^cde^	215.3 ± 8.6 ^hi^	0.097 ± 0.0011 ^g^
			70	2.45 ± 0.22 ^cd^	0.985 ± 0.018 ^ij^	298.8 ± 5.4 ^de^	287.0 ± 9.4 ^c^	0.218 ± 0.0015 ^a^
			80	4.04 ± 0.34 ^cd^	1.152 ± 0.020 ^d^	317.4 ± 6.5 ^cde^	256.1 ± 14.6 ^ef^	0.198 ± 0.0011 ^b^
	1.5 m·s^−1^		60	2.53 ± 0.33 ^cd^	1.075 ± 0.003 ^ef^	299.3 ± 16.0 ^de^	235.5 ± 6.4 ^fg^	0.210 ± 0.0065 ^b^
			70	7.65 ± 0.27 ^cd^	1.281 ± 0.028 ^b^	352.7 ± 14.7 ^ab^	242.8 ± 12.5 ^fg^	0.101 ± 0.0025 ^g^
			80	22.54 ± 1.28 ^b^	1.202 ± 0.003 ^c^	331.8 ± 15.6 ^bc^	248.5 ± 5.7 ^ef^	0.124 ± 0.0010 ^de^
Frozen dried	0.5 m·s^−1^		60	<0.25 ^d^	0.994 ± 0.001 ^ij^	290.0 ± 15.0 ^e^	206.2 ± 16.7 ^i^	0.135 ± 0.0055 ^d^
			70	<0.25^d^	1.080 ± 0.002 ^ef^	302.1 ± 20.1 ^cde^	205.3 ± 5.9 ^i^	0.095 ± 0.0010 ^g^
			80	8.87 ± 0.90 ^c^	1.011 ± 0.006 ^hi^	375.5 ± 3.2 ^a^	276.0 ± 11.7 ^cd^	0.084 ± 0.0012 ^h^
	1.5 m·s^−1^		60	<0.25 ^d^	0.976 ± 0.019 ^j^	322.4 ± 7.0 ^bcd^	227.1 ± 12.2 ^gh^	0.182 ± 0.0075 ^c^
			70	6.58 ± 0.46 ^cd^	1.029± 0.015 ^h^	361.7 ± 3.5 ^a^	238.3 ± 16.6 ^fg^	0.111 ± 0.0043 ^f^
			80	27.46 ± 1.12 ^b^	1.067 ± 0.019 ^f^	331.1 ± 9.3 ^bc^	263.9 ± 3.4 ^de^	0.124 ± 0.0077 ^ef^
Freeze dried				115.48 ± 2.29 ^a^	1.103 ± 0.019 ^e^	327.8 ± 1.24 ^cde^	410.4 ± 9.4 ^b^	0.064 ± 0.0001 ^i^
Fresh raspberry				118.27 ± 2.88 ^a^	1.635 ± 0.025 ^a^	386.1 ± 21.1 ^a^	511.7 ± 5.0 ^a^	0.053 ± 0.0005 ^j^

* Different lowercase letters indicate significant differences (*p* < 0.05).

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
