# Peer review of "Convective Drying of Fresh and Frozen Raspberries and Change of Their Physical and Nutritive Properties"

_foods, 2019, doi:10.3390/foods8070251_

Round 1

Reviewer 1 Report

Authors described the affect of freeze-drying in the quality parameters of raspberry. The manuscript is really intresting. 

Below are my comments

- In the introduction section certain antyocyanins and carotenoids present in raspberry fruit should be mentioned.

- Methods: Drying performed in dark condition? This should be mentioned in the method. Light can affect the stability of carotenoids.

-Line 149: Antioxidant activity should replaced with radical scavenging as its a general term that should be used when several techniques performed. 

Author Response

Responce to reviewer 1:

On behalf of our research team, I would like to thank the Referee #1 for reviewing our manuscript, affirming it sclarity and comprehensibility, and for her/his suggestions on how to improve the manuscript. To the best of our knowledge we have tried to respond to all raised concerns. We hope that you will find our responses satisfactorily.

Point 1. - In the introduction section certain antyocyanins and carotenoids present in raspberry fruit should be mentioned.

Responce1. As suggested, we include certain antyocyanins and carotenoids present in raspberry fruit (lines 40-41, 43).

Point 2.- Methods: Drying performed in dark condition? This should be mentioned in the method. Light can affect the stability of carotenoids.

Responce 2. As suggested, we added additional explanation: „The dryer chamber door was made up of glass, hence light could have some minimal impact on the quality of drying products.“ (lines110-111)

Ponit 3.-Line 149: Antioxidant activity should replaced with radical scavenging as its a general term that should be used when several techniques performed.

Responce 3. Changes were made. (lines 154, 177-183, 358, 370-384, 390-391, 397-398)

Reviewer 2 Report

Manuscript ID: foods-533615

Article Title: Convective drying of fresh and frozen raspberries and  change of their physical and nutritive properties

Journal: Foods

This is an interesting report on looking for a ways preserve Raspberries fresh fruits. Nowadays there is a growing interest in red fruits in general, namely raspberries. However, not always is profitable their commercialization by the existence of many suppliers. So is important to get alternative forms of valorization the fruits.  Drying is one methodology used for their preservation, whereas freeze drying usually lead to better results of a nutritive pint of view. Nevertheless, convective drying is also an option for drying.  

Regarding manuscript, the introduction is comprehensive, well stated and quite complete with up to date bibliographic references. The experimental work was well designed, and evaluates the main interesting properties of raspberries. Bibliography was used in order to compare the obtained results. In general, the conclusions are in accordance with the data obtained.

Considering this, I still have some suggests and questions:

-          2.7.4. Determination of Antioxidant activity (AOA)

Usually the volume of DPPH reagent is higher than the volume of sample. In this case, 25 mL of diluted sample was added to 90 μL of DPPH. Is this correct? How it was performed the blank experiment?

 In my opinion the used ratio DPPH/ sample may lead to erroneous results by dilution effect of DPPH reagent.

In the paper used as reference (Yen, G.-C.; Chen, H.-Y. Antioxidant activity of various tea extracts in relation to their antimutagenicity. J.  Agr. Food Chem. 1995, 43, 27-32, doi:10.1021/jf00049a007.) it was used 4 mL of sample extract to 1 mL of DPPH reagent.

-          Results

There are a lack of discussion of the importance of time needed to the different drying process studied.

-          4.  Conclusions

Nevertheless is assumed that convective drying as an process that implies lower cost comparing to freeze drying, once no economic data were presented along the manuscript, in the  final sentence the part of “…as  sufficientand  economic  alternative… “ should be removed or said   of a different way.

Author Response

Responce to reviewer 2:

On behalf of our research team, I would like to thank the Referee #2 for the interest and helpful comments that will greatly improve the clarity of the manuscript. To the best of our knowledge we have tried to respond to all raised concerns. We hope that you will find our responses satisfactorily.

Point 1. 2.7.4. Determination of Antioxidant activity (AOA)

Usually the volume of DPPH reagent is higher than the volume of sample. In this case, 25 mL of diluted sample was added to 90 μL of DPPH. Is this correct? How it was performed the blank experiment?

 In my opinion the used ratio DPPH/ sample may lead to erroneous results by dilution effect of DPPH reagent.

In the paper used as reference (Yen, G.-C.; Chen, H.-Y. Antioxidant activity of various tea extracts in relation to their antimutagenicity. J.  Agr. Food Chem. 1995, 43, 27-32, doi:10.1021/jf00049a007.) it was used 4 mL of sample extract to 1 mL of DPPH reagent.

Responce1. We apologize for our mistakes with the references, in the revised manuscript we made all necessary changes (lines 177-183)

Point 2. - Results

There are a lack of discussion of the importance of time needed to the different drying process studied.

Responce2. As requested, we included explanation in the R&D: “During the conventional air-drying, increasing drying temperatures reduce the drying time whereas shorter drying times may result in reduced risks of food quality deterioration[1].” (lines 298-301)

Point 3. - 4.  Conclusions

Nevertheless is assumed that convective drying as an process that implies lower cost comparing to freeze drying, once no economic data were presented along the manuscript, in the  final sentence the part of “…as  sufficientand  economic  alternative… “ should be removed or said   of a different way.

Responce3. Changes were made. Moreover, we highlighted the economic issues of freeze drying as compared to conventional drying (line 54-63)

Reviewer 3 Report

The manuscript shows interesting results worth of publication. However, this manuscript requires improving due to many shortcomings.

Comments that should be considered to make the manuscript suitable for publication:

L 25. Separate ”hadless” into “had less”.

L 92. Separate ”raspberriesat” into “raspberries at”.

L 104. Separate ”inprocessing” into in processing”.

L 117. The constant moisture content is known as equilibrium moisture content which depends on the temperature of hot air used for drying. In the experiment there were three temperature levels of the hot air. It is not clear why only one value was presented and which temperature this value concerns. If this value was not a equilibrium moisture content it is better to write that the samples were dried until the same value of moisture content.

L 119-121. I recommend to describe the statistical part in a separate chapter at the very end of methodological section as ANOVA was performed also for other tests.

L 124-125. “For drying regime a sample of raspberries (n=45) gave shrinkage before and after convective drying…” This sentence requires editing as it directly concerns determination of fruit volume before and after drying.

L 208. In fact the results obtained by the Authors within this study are not consistent with the results obtained by Pavkov et al. 2017. Please try to discuss the results obtained carefully. Note, that the range of temperatures was different comparing to the relevant range in the cited reference.

L 234. Delete dot after Figure 2.

L 240-242. Provide standard deviations to particular dimensions values a, b and c. Please check, whether the picture of convective dried fruit is correct. The dimensions for this fruit should be much lower assuming that volume shrinkage for 60C exceeded 30%.

L 256. Why the values of color parameters for fresh samples are different for particular drying conditions? If there were mean values please provide standard deviation.

L 257. It is not clear what "negative influences" mean? Consider using “significant influence” or “considerable impact”.

L 258. Consider to explain the color change of freeze dried samples taking into account the considerable increase in lightness.

L 291. Explain a method for measuring the temperature of samples.

L 299. Refer to Table 4 when discussing the content of ascorbic acid.

L 320-322. How to explain the higher content of total phenols in hot air dried fresh samples compared to freeze dried samples (Table 4)?

L 342-352.  The discussion on antioxidant activity does not match the data in the Table 4 as the highest values were obtained for fresh convective dried samples at 70C (0.5 m/s) and 60C (1.5 m/s). Please comment on this.

L 358-39.  In the conclusions there are missed results regarding biologically active compounds and antioxidant activity.

Author Response

Responce to reviewer 3:

On behalf of our research team, I would like to thank the Referee #3 for the interest and helpful comments that will greatly improve the clarity of the manuscript. To the best of our knowledge we have tried to respond to all raised concerns. We hope that you will find our responses satisfactorily.

Point 1.L 25. Separate ”hadless” into “had less”.

Responce1. It seems that some formatting problems occurred during the submission, however, in the revised manuscript this problem has been resolved.(line 25)

Point 2.L 92. Separate ”raspberriesat” into “raspberries at”.

Responce 2. Changed (line 98)

Point 3.L 104. Separate ”inprocessing” into ”in processing”.

Responce 3. Changed (line 109-110)

Point 4.L 117. The constant moisture content is known as equilibrium moisture content which depends on the temperature of hot air used for drying. In the experiment there were three temperature levels of the hot air. It is not clear why only one value was presented and which temperature this value concerns. If this value was not a equilibrium moisture content it is better to write that the samples were dried until the same value of moisture content.

Responce 4.  We thank the reviewer for this comment, which is very useful for improving the Manuscript. All samples were drieduntil the same value of moisture content. (lines 124-125)

Point 5.L 119-121. I recommend to describe the statistical part in a separate chapter at the very end of methodological section as ANOVA was performed also for other tests.

Responce 5.  We added a new subsection 2.8. Statistical analysis. (lines 210-213)

Point 6.L 124-125. “For drying regime a sample of raspberries (n=45) gave shrinkage before and after convective drying…” This sentence requires editing as it directly concerns determination of fruit volume before and after drying.

Responce 6. Changed (lines 129-131)

Point 7. L 208. In fact the results obtained by the Authors within this study are not consistent with the results obtained by Pavkov et al. 2017. Please try to discuss the results obtained carefully. Note, that the range of temperatures was different comparing to the relevant range in the cited reference.

Responce7. We accept this suggestion and made requested changes. (lines 220-229)

Point 8.L 234. Delete dot after Figure 2.

Responce 8. Corrected (250)

Point 9.L 240-242. Provide standard deviations to particular dimensions values a, b and c. Please check, whether the picture of convective dried fruit is correct. The dimensions for this fruit should be much lower assuming that volume shrinkage for 60C exceeded 30%.

Responce 9. As requested, we added SD to particular dimensions values a, b and c and also correct the mean values as some random mistakes when writing results were occurred. (lines 256-259)

 Point 10.L 256. Why the values of color parameters for fresh samples are different for particular drying conditions? If there were mean values please provide standard deviation.

Responce 10. SD for colour measurements were not provided in Table 2 as in such case, this addition will overload the space of the table. However, all SD are under 5% value from their mean values.

The differences between colour parameters with respect to fresh and dried samples are given via colour changes ΔE (Table 2) where clearly can be seen whereas colour changes are significant or not.

Point 11.L 257. It is not clear what "negative influences" mean? Consider using “significant influence” or “considerable impact”.

Responce 11. Changed (lines 23, 274, 369)

Point 12.L 258. Consider to explain the color change of freeze dried samples taking into account the considerable increase in lightness.

Responce 12. We added an explanation (lines 276-277)

Point 13.L 291. Explain a method for measuring the temperature of samples.

Responce 13. We included additional explanation (lines 314-315)

Point 14.L 299. Refer to Table 4 when discussing the content of ascorbic acid.

Responce 14. Changed (line 309, 323)

Point 15.L 320-322. How to explain the higher content of total phenols in hot air dried fresh samples compared to freeze dried samples (Table 4)?

Responce 15. We included additional explanation (lines 343-351)

Point 16.L 342-352.  The discussion on antioxidant activity does not match the data in the Table 4 as the highest values were obtained for fresh convective dried samples at 70C (0.5 m/s) and 60C (1.5 m/s). Please comment on this.

Responce 16. We included additional explanation (lines 371-374; 376-378)

Point 17.L 358-39.  In the conclusions there are missed results regarding biologically active compounds and antioxidant activity.

Responce 17. We included additional explanation (lines 395-398)

Reviewer 4 Report

A well-written study on convective drying of raspberries. Authors compare results to freeze drying, the de-facto standard. Variables of investigation are of practical interest (colour, shape, ingredients). Authors should further stress why their convective drying was better compared to the previous studies on convective drying (process desing? operation regime?). To further strengthen the economic argument, give a cost estimate for drying of some amount by either method.

Minor: 

Please check: In the PDF sometimes blank spaces are missing. Is it an artifact of the conversion?

Please change the symbol in eq 2., V is a relative volume.

Figures: Remove the green background.

Author Response

Responce to reviewer 4:

On behalf of our research team, I would like to thank the Referee # for the interest and helpful comments that will greatly improve the clarity of the manuscript. To the best of our knowledge we have tried to respond to all raised concerns. We hope that you will find our responses satisfactorily.

Point 1.Please check: In the PDF sometimes blank spaces are missing. Is it an artifact of the conversion?

Responce1. It seems that some formatting problems occurred during the submission, however, in the revised manuscript this problem has been resolved.

Point 2.Please change the symbol in eq 2., V is a relative volume.

Responce 2. As suggested, we changed this part(lines 134-135).

Point 3.Figures: Remove the green background.

Responce 3. Removed

Round 2

Reviewer 3 Report

The revised version of the manuscript has been significantly improved by the authors.